# Diffusion Models for Contrast Harmonization of Magnetic Resonance Images

**Alicia Durrer**[1]                   ALICIA.DURRER@UNIBAS.CH

**Julia Wolleb**[1]                   JULIA.WOLLEB@UNIBAS.CH

**Florentin Bieder**[1]                FLORENTIN.BIEDER@UNIBAS.CH

**Tim Sinnecker**[1,2]                 TIM.SINNECKER@USB.CH

**Matthias Weigel**[1,2]               MATTHIAS.WEIGEL@UNIBAS.CH

**Robin Sandkühler**[1]             ROBIN.SANDKUEHLER@UNIBAS.CH

**Cristina Granziera**[1,2]             CRISTINA.GRANZIERA@USB.CH

**Özgür Yaldizli**[1,2]               OEZGUER.YALDIZLI@USB.CH

**Philippe C. Cattin**[1]              PHILIPPE.CATTIN@UNIBAS.CH

[1] *Department of Biomedical Engineering, University of Basel, Allschwil, Switzerland*

[2] *University Hospital Basel, Switzerland*

**Editors:** Accepted for publication at MIDL 2023

## Abstract

Magnetic resonance (MR) images from multiple sources often show differences in image contrast related to acquisition settings or the used scanner type. For long-term studies, longitudinal comparability is essential but can be impaired by these contrast differences, leading to biased results when using automated evaluation tools. This study presents a diffusion model-based approach for contrast harmonization. We use a data set consisting of scans of 18 Multiple Sclerosis patients and 22 healthy controls. Each subject was scanned in two MR scanners of different magnetic field strengths ($1.5\,\mathrm{T}$ and $3\,\mathrm{T}$), resulting in a paired data set that shows scanner-inherent differences. We map images from the source contrast to the target contrast for both directions, from $3\,\mathrm{T}$ to $1.5\,\mathrm{T}$ and from $1.5\,\mathrm{T}$ to $3\,\mathrm{T}$. As we only want to change the contrast, not the anatomical information, our method uses the original image to guide the image-to-image translation process by adding structural information. The aim is that the mapped scans display increased comparability with scans of the target contrast for downstream tasks. We evaluate this method for the task of segmentation of cerebrospinal fluid, grey matter and white matter. Our method achieves good and consistent results for both directions of the mapping.

**Keywords:** Diffusion models, contrast harmonization, image-to-image translation

## 1. Introduction

In medical studies using magnetic resonance (MR) images, data acquisition from multiple centers and different scanners is a common scenario, especially regarding comprehensive or long-term studies (Krüger et al., 2020). However, challenges arise when we compare data acquired with different MR scanners since the obtained MR images often display differences related to the acquisition settings and scanner variability (Dadar et al., 2020). For instance, as the magnetic field strength alters the level of contrast between different tissue types (Maubon et al., 1999; Ba-Ssalamah et al., 2003), the location of borders between different tissues can vary in images acquired with MR scanners of different field strength

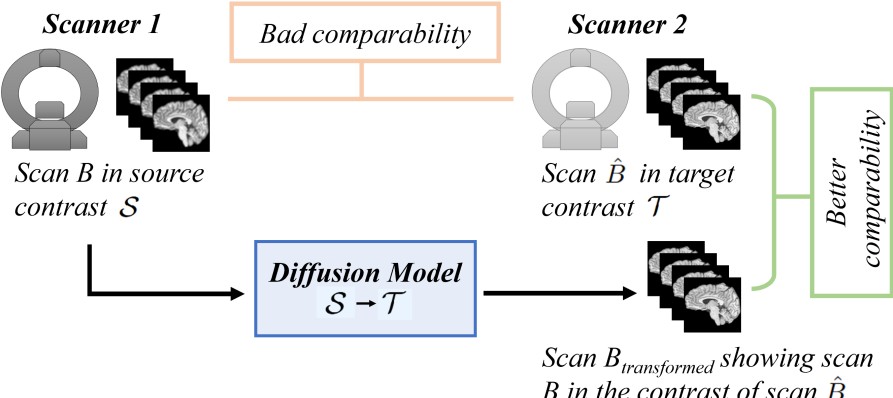

Figure 1: Overview of our contrast harmonization method. We train a diffusion model using paired data from source contrast $\mathcal{S}$ and target contrast $\mathcal{T}$. We translate scan $B \in \mathcal{S}$ to scan $B_{transformed}$ that appears in contrast $\mathcal{T}$, allowing better comparability with $\hat{B} \in \mathcal{T}$ in subsequent tasks, such as segmentation.

(Keihaninejad et al., 2010). Therefore, direct comparison of scans from different scanners with automated tools such as SIENA (Smith et al., 2002, 2004) is highly difficult. Consequently, a scanner change in the middle of a long-term study affects the automatic evaluation and longitudinal comparability of scans (Sinnecker et al., 2022). Longitudinal studies are important to monitor progressive diseases such as Multiple Sclerosis (MS), a demyelinating central nervous system disease (Mahad et al., 2015). It is essential that subsequent scans allow reliable interpretation of the disease progression without a bias related to a scanner change between the image acquisitions. In this work, we focus on a scanner change from $1.5\,\mathrm{T}$ to $3\,\mathrm{T}$ magnetic field strength that took place within a longitudinal MS study (Disanto et al., 2016). The goal of this work is to restore the comparability of the images by mapping all images to the same target contrast. A data set was acquired by (Sinnecker et al., 2022), for which healthy subjects as well as MS patients were scanned in the $1.5\,\mathrm{T}$ and the $3\,\mathrm{T}$ scanner within approximately 3.5 months. Due to the short time span between the acquisitions, we assume that the differences in the images of the same participant are only related to the different scanner types. We therefore have paired $1.5\,\mathrm{T}$ and $3\,\mathrm{T}$ data for each participant.

**Contribution** By adapting a Denoising Diffusion Probabilistic Model (DDPM) (Ho et al., 2020; Wolleb et al., 2022b) for contrast harmonization, we translate images from a source contrast $\mathcal{S}$ to a target contrast $\mathcal{T}$. Considering a pair $B \in \mathcal{S}$, $\hat{B} \in \mathcal{T}$, we map scan $B$ slice-by-slice to scan $B_{transformed}$, appearing in the contrast of $\mathcal{T}$, as shown in Figure 1. We generate consistent three-dimensional (3D) volumes by stacking two-dimensional (2D) slices, allowing us to save memory during image-to-image translation. Compared to the original $B$, $B_{transformed}$ presents better comparability with $\hat{B}$ from $\mathcal{T}$, with respect to downstream tasks such as segmentation of grey matter (GM), white matter (WM) and cerebrospinal fluid (CSF) using FAST (Zhang et al., 2001). We achieve good results for both directions

of the mapping, i.e., from $3\,\mathrm{T}$ to $1.5\,\mathrm{T}$ and from $1.5\,\mathrm{T}$ to $3\,\mathrm{T}$. Our code is available at https://github.com/AliciaDurrer/dm_moni_mr.

**Related Work**  Tracking brain volume changes over different scanners is prone to an error related to the scanner change (Lee et al., 2019). (Sinnecker et al., 2022) showed that given a paired data set, a corrective term for the volume computation can be calculated. To achieve a higher level of generalizability, images should be mapped to the target contrast. DeepHarmony (Dewey et al., 2019), builds on the U-Net architecture (Ronneberger et al., 2015) and was developed for MR image contrast harmonization across protocol or scanner changes. For almost a decade, GANs (Goodfellow et al., 2014) have been the state of the art for image generation and image-to-image translation (Karras et al., 2021; Emami et al., 2020; Zhu et al., 2017; Isola et al., 2017). GANs such as (Dar et al., 2019; Liu et al., 2020; Luo et al., 2021; Peng et al., 2021) were created for multi-modal MR image synthesis, for instance T1- to T2-contrast. (Nie et al., 2017, 2018) performed cross-modal and cross-scanner image synthesis using GANs. Lately, DDPMs (Sohl-Dickstein et al., 2015; Ho et al., 2020) became the focus of attention. (Nichol and Dhariwal, 2021; Dhariwal and Nichol, 2021) further improved DDPMs, resulting in a transition of the state of the art for image generation from GANs to diffusion models. Their application includes text-to-image generation as in (Rombach et al., 2022; Saharia et al., 2022b), image-to-image translation (Saharia et al., 2022a; Seo et al., 2022; Wolleb et al., 2022c), inpainting (Saharia et al., 2022a; Lugmayr et al., 2022; Wolleb et al., 2022c) and deformable image registration (Kim et al., 2022). The recently introduced diffusion models are also used in medical image analysis. For instance for cross-modal (Lyu and Wang, 2022) and multi-modal MR image synthesis (Özbey et al., 2022), anomaly detection (Wolleb et al., 2022a) and synthetic image generation (Pinaya et al., 2022). In medical studies, processing of 3D data is often required. (Dorjsembe et al., 2022) showed the applicability of diffusion models to 3D data, but frequently, memory restrictions affect the processing of large volumes.

## 2. Method

DDPMs as described in (Nichol and Dhariwal, 2021) form the basis for the proposed method. They are a class of generative models based on an iterative noising process $q$ and denosing process $p_\theta$. In the forward process $q$, Gaussian noise is added to an input image $x$ for $T$ time steps $t$. As the noise level is increased from a minimum at $t = 0$ to a maximum at $t = T$, each image $x_0, x_1, ..., x_T$ displays a higher amount of noise compared to the previous one. The forward noising process $q$ is defined as

$$q(x_t|x_{t-1}) := \mathcal{N}(x_t; \sqrt{1 - \beta_t}x_{t-1}, \beta_t\mathbf{I}), \tag{1}$$

where $\mathbf{I}$ is the identity matrix and $\beta_1, ..., \beta_T$ are the forward process variances. With $\alpha_t := 1 - \beta_t$ and $\overline{\alpha}_t := \prod_{s=1}^{t} \alpha_s$ and using the reparametrization trick, $x_t$ can be written as

$$x_t = \sqrt{\overline{\alpha}_t}x_0 + \sqrt{1 - \overline{\alpha}_t}\epsilon, \quad \text{with } \epsilon \sim \mathcal{N}(0, \mathbf{I}). \tag{2}$$

For the denoising process $p_\theta$, the aim is to reverse the forward process, hence to predict $x_{t-1}$ from $x_t$ for $t \in \{T, ..., 1\}$. The learned model parameters $\theta$ define the reverse process

$$p_\theta(x_{t-1}|x_t) := \mathcal{N}\big(x_{t-1}; \mu_\theta(x_t, t), \Sigma_\theta(x_t, t)\big). \tag{3}$$

To map images of the source contrast $\mathcal{S}$ to the target contrast $\mathcal{T}$, we use the generative process of DDPMs. We adapt the method of (Wolleb et al., 2022b), originally created for DDPM-based image segmentation using paired data, to translate an image $B \in \mathcal{S}$ to $B_{transformed}$, which appears in the target contrast $\mathcal{T}$. The used data set provides for each of the $m$ participants one 3D scan $B$ of the source contrast $\mathcal{S}$ and one corresponding 3D scan $\hat{B}$ of the target contrast $\mathcal{T}$. Due to memory restrictions, we implement our model in 2D and slice each of the $m$ scans of both contrasts into $n$ slices and obtain for each participant $B = \{b_i\}_{i=1}^n$ and $\hat{B} = \{\hat{b}_i\}_{i=1}^n$ slices. We translate the slices $\{b_i\}_{i=1}^n$ originating from a scan $B \in \mathcal{S}$ such that they resemble the contrast of the slices $\{\hat{b}_i\}_{i=1}^n$ originating from $\hat{B} \in \mathcal{T}$, whereby each baseline slice $b_i$ has its corresponding ground truth $\hat{b}_i$.

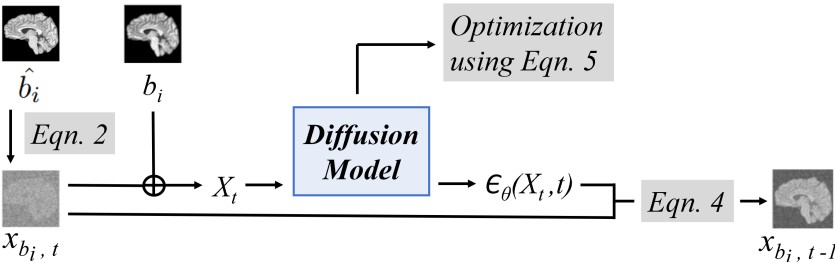

Figure 2: Overview of the training. Anatomical information is given through the concatenation of image $b_i$ from $B \in \mathcal{S}$ with noisy image $x_{b_i,t}$. $X_t$ is used by the diffusion model to predict a slightly denoised image $x_{b_i,t-1}$ from $x_{b_i,t}$ using Equation (4).

During training, depicted in Figure 2, we pick a random timestep $t \in \{1,...,T\}$ and apply Equation (2) with $x_0 = \hat{b}_i$ to compute a noisy image $x_{b_i,t}$ from $\hat{b}_i$. Since we only want to change the scanner-related image contrast and not any anatomical features, we add anatomical information of our baseline image $b_i$ through concatenation. We define the concatenated image as $X_t := b_i \oplus x_{b_i,t}$ which serves as input for our diffusion model. We can compute $x_{b_i,t-1}$ using Equation (4), which summarizes a denoising step as

$$x_{b_i,t-1} = \frac{1}{\sqrt{\alpha_t}}\left(x_{b_i,t} - \frac{1-\alpha_t}{\sqrt{1-\overline{\alpha}_t}}\epsilon_\theta(X_t,t)\right) + \sigma_t \mathbf{z}, \quad \text{with } \mathbf{z} \sim \mathcal{N}(0,\mathbf{I}), \quad (4)$$

whereby $\epsilon_\theta(X_t,t)$ is the output of the diffusion model at time step $t$, $\sigma_t$ describes the variance scheme and $\mathbf{z}$ denotes the stochastic component of the process. The loss used to train the diffusion model $\epsilon_\theta$ can be written as

$$|| \epsilon - \epsilon_\theta(X_t,t) ||^2 = || \epsilon - \epsilon_\theta(b_i \oplus (\sqrt{\bar{a}_t}\hat{b}_i + \sqrt{(1-\bar{a}_t)}\epsilon),t) ||^2, \quad \text{with } \epsilon \sim \mathcal{N}(0,\mathbf{I}). \quad (5)$$

For the two directions of the image mapping, $3\,\mathrm{T}$ to $1.5\,\mathrm{T}$ and $1.5\,\mathrm{T}$ to $3\,\mathrm{T}$, two separate models need to be trained, as source and target contrast change. To translate a scan volume $B = \{b_i\}_{i=1}^n \in \mathcal{S}$ to the target contrast $\mathcal{T}$ during sampling, we translate every slice $b_i$ to the synthetic slice $x_{b_i,0}$ in the contrast of $\mathcal{T}$. Figure 3 summarizes the sampling process

starting from $x_{b_i,T} \sim \mathcal{N}(0, \mathbf{I})$. The previously trained denoising model is now applied for every denoising step $t \in \{T, ..., 1\}$ using Equation (4). Anatomical information is also added for every step $t$ of the denoising process through the concatenation of $b_i$ and $x_{b_i,t}$. We then stack the $n$ output slices $x_{b_i,0}$ to create our final 3D output volume $B_{transformed} = \{x_{b_i,0}\}_{i=1}^n$, displaying the whole brain in target contrast.

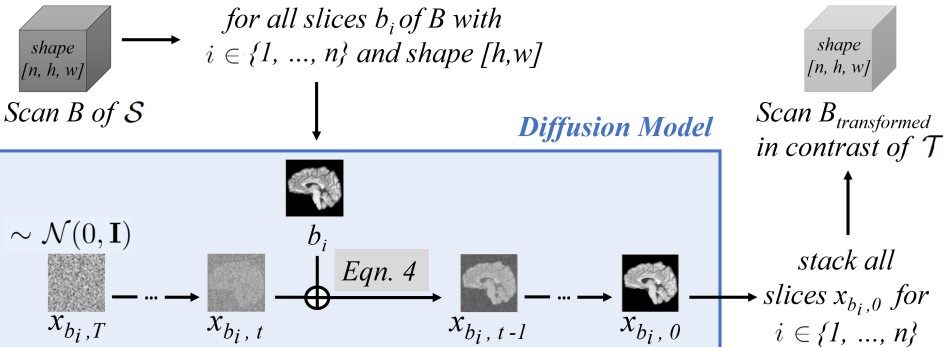

Figure 3: Translation from $B \in \mathcal{S}$ to $B_{transformed}$. Each slice $b_i$ of $B \in \mathcal{S}$ is iteratively denoised by applying Equation (4) for steps $t \in \{T, ..., 1\}$, whereby slice $b_i$ is used to add anatomical information through concatenation. The 2D output slices $\{x_{b_i,0}\}_{i=1}^n$ get stacked to $B_{transformed}$, showing the input scan $B$ translated to $\mathcal{T}$.

## 3. Data Set and Training Details

We used a data set exclusively created to track a scanner change from a 1.5 T Siemens Magnetom Avanto to a 3 T Siemens Magnetom Skyra$^{fit}$ whole-body scanner in 2016 (Sinnecker et al., 2022). Scanner details can be found in Appendix A. The participants' data is not public due to data privacy protection. Written consent was obtained from all participants. The data was coded (i.e., pseudoanonymized) at the time of the enrollement of the patients and includes scans of 22 healthy controls and 18 MS patients. All participants were scanned first in the 1.5 T scanner and after a median time interval of 3.5 months in the 3 T scanner. The relatively short time span between the scans ensures that no major disease progression happened in the MS patients, allowing a direct comparison of the scans and thereby forming a paired data set. Data set details can be found in Appendix B. Pre-processing of the original 3D data includes skull-stripping using HD-BET (Isensee et al., 2019), biasfield correction (Tustison et al., 2010), resampling of voxel spacing to 1 x 1 x 1 mm$^3$ removal of the top and bottom 0.1 percentile of voxel intensities and normalization to voxel values between 0 and 1. With the ANTsPyx package, the paired images were registered based on affine and deformable transformation, using mutual information as similarity metric and an elastic regularization (Avants et al., 2014). We sliced each pre-prepocessed 3D scan into 160 sagittal slices of shape [246, 262] that were then cropped to a size of [224, 224]. The cropping only affected background pixels. We trained our models for 300,000 iterations with a batch size of four on a NVIDIA GeForce RTX 2080 Ti GPU, taking about 60 hours.

As in (Wolleb et al., 2022b), the number of channels in the first layer of the model is 128, and one attention head is used at resolution 16, resulting in 11,402,370 model parameters. The learning rate used is $10^{-4}$ for the Adam optimizer. $T$ is set as 1000. Details on hyperparameters and architecture can be found in (Nichol and Dhariwal, 2021). In addition to MSE calculation and histogram analysis, we evaluated CSF, GM and WM segmentations using FAST (Zhang et al., 2001) for the original images and the generated images. We did a four-fold cross-validation, combining data of healthy controls and MS patients in each fold. For each fold, the slices of 30 scans were used for training and those of 10 scans for testing.

## 4. Results and Discussion

For the evaluation, we compare our diffusion model ($DM$) with DeepHarmony ($DH$) (Dewey et al., 2019) and $pGAN$ (Dar et al., 2019). Implementation details of the comparing methods can be found in Appendix C. Each method takes an image $B \in \mathcal{S}$ and generates the image $B_{transformed}$ appearing in target contrast. The direct comparison of $B$ versus $\hat{B}$ is denoted as *Original*.

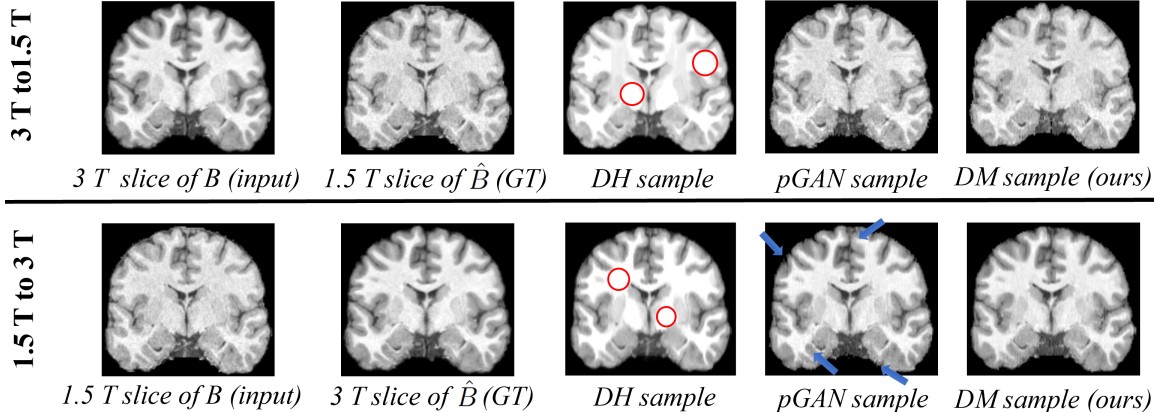

Figure 4: An exemplary coronal slice of a scan $B \in \mathcal{S}$, the corresponding ground truth (GT) slice of $\hat{B} \in \mathcal{T}$ and slices of its mappings $B_{transformed}$ in contrast $\mathcal{T}$ generated by $DH$, $pGAN$ and our $DM$ are shown for both mapping directions. The red circles indicate hyperintense regions generated by $DH$. The blue arrows point at stripe artifacts produced by $pGAN$. Further examples are provided in Appendix F.

Figure 4 contains examples of generated images and the corresponding input and ground truth slices. The original 1.5 T and 3 T images differ in contrast. While $pGAN$ and our $DM$ generate convincing mappings of the input images to target contrast, $DH$ blurs the image and increases the brightness excessively for both directions of the image mapping. The examples shown are coronal slices from a 3D volume that was built by stacking sagittal slices. While images mapped to 3 T contrast by $pGAN$ contain some stripe artifacts, mostly in the border regions of the brain, our $DM$ does not create any stripe artifacts, indicating that processing the data in a slice-wise fashion is a valuable simplification.

Table 1: MSE and AHD scores (formulas in Appendix D) for both directions of the mapping. This table shows the average scores on the test set over a four-fold cross-validation, the variances are listed in Appendix D. The best results are bold.

| | 3 T to 1.5 T | | 1.5 T to 3 T | |
| Method | MSE | AHD | MSE | AHD |
|---|---|---|---|---|
| *Original* | $1.85 \times 10^{-3}$ | $6.15 \times 10^5$ | $1.85 \times 10^{-3}$ | $6.15 \times 10^5$ |
| *DH* | $2.30 \times 10^{-3}$ | $7.49 \times 10^5$ | $1.73 \times 10^{-3}$ | $8.22 \times 10^5$ |
| *pGAN* | $\mathbf{1.50 \times 10^{-3}}$ | $\mathbf{1.27 \times 10^5}$ | $0.94 \times 10^{-3}$ | $2.07 \times 10^5$ |
| *DM* (ours) | $1.61 \times 10^{-3}$ | $1.31 \times 10^5$ | $\mathbf{0.76 \times 10^{-3}}$ | $\mathbf{1.48 \times 10^5}$ |

To compare the different methods, we compute the Mean Squared Error (MSE) between the ground truth images $\hat{B}$ and the translated images $B_{transformed}$. In Table 1 we report the MSE as well as the sum of the bin-wise absolute difference between the histograms (AHD), whereby each histogram consists of 255 bins. Mapping images of 3 T to 1.5 T contrast, our *DM* and the *pGAN* manage to decrease the MSE and to improve the AHD compared to *Original*. *DH* cannot compete and produces a higher MSE and AHD than the *Original*. Mapping 1.5 T to 3 T contrast, our *DM* outperforms *Original*, *DH* and *pGAN* regarding MSE and AHD. The results show that the contrast harmonization shifts the voxel distributions characteristic for $\mathcal{S}$ towards the distributions of $\mathcal{T}$. For both directions of the mapping, exemplary histograms can be found in Appendix E. The histograms of the samples generated by *pGAN* and our *DM* better align with the histogram of the ground truth $\hat{B} \in \mathcal{T}$ than the histogram of $B \in \mathcal{S}$, for both directions of the mapping.

Table 2: Absolute volume differences in mm$^3$ of the segmentations of CSF, GM and WM of all original images $B \in \mathcal{S}$ and generated images $B_{transformed}$ compared to ground truths $\hat{B} \in \mathcal{T}$ for both directions of the mapping. This table shows the average scores on the test set over a four-fold cross-validation, the variances are listed in Appendix D. The best result per class is bold.

| | 3 T to 1.5 T | | | 1.5 T to 3 T | | |
| | Absolute Volume Difference | | | Absolute Volume Difference | | |
| Method | CSF | GM | WM | CSF | GM | WM |
|---|---|---|---|---|---|---|
| *Original* | $3.72 \times 10^4$ | $9.32 \times 10^4$ | $8.88 \times 10^4$ | $3.72 \times 10^4$ | $9.32 \times 10^4$ | $8.88 \times 10^4$ |
| *pGAN* | $\mathbf{1.12 \times 10^4}$ | $\mathbf{1.14 \times 10^4}$ | $1.91 \times 10^4$ | $3.11 \times 10^4$ | $6.15 \times 10^4$ | $2.09 \times 10^4$ |
| *DM* (ours) | $1.95 \times 10^4$ | $1.54 \times 10^4$ | $\mathbf{1.59 \times 10^4}$ | $\mathbf{1.79 \times 10^4}$ | $\mathbf{3.91 \times 10^4}$ | $\mathbf{1.49 \times 10^4}$ |

To obtain further insight about the increased comparability of $B_{transformed}$ with $\hat{B} \in \mathcal{T}$ for downstream tasks, we segmented the original 3D images as well as the 3D images generated by our *DM* and *pGAN* into the three classes CSF, GM and WM using FAST (Zhang

et al., 2001). *DH* could not be considered for this comparison, as the quality of the generated images was not high enough to create meaningful segmentations using FAST. Examples of the segmentations can be found in Appendix G. As the voxel size is 1 x 1 x 1 mm$^3$, we calculate the volume of each class by counting the voxels attributed to each class. For both directions of the image mapping we then compute the volume differences for each class between the ground truth segmentation of $\hat{B}$ and the segmentations of $B$ or $B_{transformed}$, respectively. Table 2 shows that *pGAN* and our *DM* decrease the differences between the segmentation volumes of $B_{transformed}$ and $\hat{B}$ compared to the segmentation volume difference between $B$ and $\hat{B}$ for both directions of the mapping. *pGAN* and our *DM* show similar performance for the mapping from 3 T to 1.5 T but our model performs better for the mapping from 1.5 T to 3 T. We conclude that harmonizing contrasts before segmenting allows more coherent assignment of voxels to classes, enabling better comparison of tissue volumes between scans. We use the Dice score and the Hausdorff distance (HD) to assess that the contrast harmonization did not negatively affect the location of the segmented volumes, whereby the ground truth is given by the segmentation of $\hat{B}$. Table 3 shows that our *DM* achieves better HD scores than *pGAN* and the *Original* for all classes. The Dice scores for our *DM* remain in the ranges of the *Original* for the mapping from 3 T to 1.5 T. For the mapping from 1.5 T to 3 T, however, our *DM* improves the Dice scores of GM and WM considerably compared to the *Original*. According to Table 1 and Table 2, *pGAN* performed better than our *DM* for the mapping from 3 T to 1.5 T, but regarding the HD, our *DM* seems more reliable for both directions of the mapping. The results indicate that using our *DM*, the contrast harmonization and the resulting change in voxel distribution and segmentation volumes occurs at the desired regions.

Table 3: Dice scores and HD of the segmentations of CSF, GM and WM of all original images $B \in \mathcal{S}$ and generated images $B_{transformed}$ compared to the ground truth segmentations of $\hat{B} \in \mathcal{T}$ for both directions of the mapping. This table shows the average scores on the test set over a four-fold cross-validation, the variances are listed in Appendix D. The best result per class is bold.

| | 3 T to 1.5 T | | | | | | 1.5 T to 3 T | | | | | |
| | Dice | | | HD | | | Dice | | | HD | | |
| Method | CSF | GM | WM | CSF | GM | WM | CSF | GM | WM | CSF | GM | WM |
|---|---|---|---|---|---|---|---|---|---|---|---|---|
| *Original* | 0.82 | **0.82** | 0.87 | 10.21 | 8.68 | 11.16 | **0.82** | 0.82 | 0.87 | 10.21 | 8.68 | 11.16 |
| *pGAN* | **0.82** | 0.79 | 0.87 | 21.48 | 13.54 | 12.14 | 0.76 | 0.85 | 0.92 | 8.67 | 7.24 | 9.48 |
| *DM* (ours) | 0.80 | 0.79 | **0.88** | **9.11** | **6.73** | **7.67** | 0.81 | **0.88** | **0.92** | **8.65** | **7.01** | **9.30** |

## 5. Conclusion

We present a novel method for contrast harmonization based on DDPMs. Using paired data from a source contrast $\mathcal{S}$ and a target contrast $\mathcal{T}$, our method allows us to translate a scan $B$ of the source contrast $\mathcal{S}$ to scan $B_{transformed}$ appearing in the contrast of $\mathcal{T}$. For the image-to-image translation, our diffusion model receives information from the source

image $B$, as we only want to adjust the contrast while keeping the anatomical information. The translation improves comparability between scans from different contrasts for further evaluation and downstream tasks such as tissue segmentation. Our model outperforms the comparing methods for the mapping from $1.5\,\mathrm{T}$ to $3\,\mathrm{T}$ and generates great results for the opposite image mapping. Compared to GANs, diffusion models are not trained in an adversarial manner, making the training straightforward. The input and output of our model are 2D slices, allowing us to save memory compared to models trained on 3D volumes. Stacking the 2D images to a 3D volume does not generate any stripe artifacts, showing us that our method only changes the contrast and not the anatomical structure. Due to the image generation characteristics of DDPMs, our method has a long sampling time compared to the other methods, which could be improved by using other sampling schemes (Song et al., 2020). So far, our model was only trained on skull-stripped data sets, limiting a more in-depth temporal analysis of brain-volume changes. As our next step, we will omit the skull-stripping during pre-processing, enabling observing brain-volume changes relative to the fixed skull size.

## Acknowledgments

This project was partially funded by a grant received by Özgür Yaldizli from the Research Fund for Junior Researchers of the University of Basel.

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

## Appendix A. Scanner Details

Table 4: MR Scanners used for the data acquisition (Sinnecker et al., 2022)

|  | **Previous Scanner** | **Current Scanner** |
|---|---|---|
| Manufacturer | Siemens Healthineers | Siemens Healthineers |
|  | Erlangen, Germany | Erlangen, Germany |
| Model Name | Magnetom Avanto | Magnetom Skyra$^{fit}$ |
| Magnetic Field Strength | 1.5 T | 3 T |
| Repetition Time | 2080 $ms$ | 2300 $ms$ |
| Inversion Time | 1100 $ms$ | 900 $ms$ |
| Echo Time | 3.1 $ms$ | 2.94 $ms$ |
| Imaging Matrix | 240 x 256 | 240 x 256 |
| Field of View (FOV) | 234 x 250 mm$^2$ | 240 x 256 mm$^2$ |
| Pixel Bandwidth | 130 $Hz$ per Pixel | 240 $Hz$ per Pixel |
| Flip Angle | 15 $degrees$ | 9 $degrees$ |
| Scanner Image Filter | Prescan-Normalization | Prescan-Normalization and |
|  |  | Distortion Correction in 3D |
| Software Version | syngo MR B17 | syngo MR VE11C |
| Sequence used for Data Set | MPRAGE | MPRAGE |

## Appendix B. Data Set Details

The MS patients are part of a longitudinal MS study (Disanto et al., 2016). Both, MS patients and healthy controls were scanned first in the 1.5 T scanner and after the scanner change in the 3 T scanner (median time interval 3.5 months, range 1.7 – 5.2 months).

Table 5: Details about the participants included in the data set, provided by (Sinnecker et al., 2022)

|  | **Multiple Sclerosis Patients** | **Healthy Controls** |
|---|---|---|
| Number of Female Participants | 14 | 11 |
| Number of Male Participants | 4 | 11 |
| Age in Years, Mean [SD] | 51.7 [±12.7] | 28.9 [±7.6] |

## Appendix C. Implementation Details

We compared our models against *DeepHarmony* (*DH*) and *pGAN*. For both comparing methods, we used the same four train and test folds as for our model to perform cross-validation.

Implementation details:

- *DH*: to make it more comparable to our method, we used 2D sagittal slices with shape [256, 256] as input, instead of the 2.5 dimensional implementation proposed in the paper. We cropped the output to [224, 224] and stacked the slices to a 3D volume of shape [160, 224, 224]. We trained the model for 300 epochs. We used a batch size of eight as in the original implementation and adjusted the learning rate to $10^{-4}$ as the original learning rate was not compatible with our images. For further implementation details refer to (Dewey et al., 2019).

- *pGAN*: we trained the models for 300 epochs (150 with normal learning rate, 150 with the learning rate decayed to zero) with a batch size of four. We used no neighbouring slices. The cycle loss weight as well as the perceptual loss weight were set to 100. The processed slices were of shape [256, 256], we cropped and stacked the output slices to shape [160, 224, 224] for comparison with our model. For further implementation details refer to (Dar et al., 2019) and https://github.com/icon-lab/pGAN-cGAN.

## Appendix D. Detailed Metrics

The MSE was calculated using

$$MSE = \sum_{i=1}^{N}(a_i - b_i)^2 \tag{6}$$

with $a$ and $b$ being the images to compare and $i$ iterating over all voxels $N$. The squared differences are summed. To calculate the AHD we used the following formula

$$AHD = \sum_{i=1}^{bins} \mid h_1(x_i) - h_2(x_i) \mid \tag{7}$$

with $h_1$ and $h_2$ being the histograms to compare for all bins $x_i$, with $i$ ranging from one to the total number of bins.

Table 6: MSE and AHD averages including variances over four-fold cross-validation for the mapping from 3 T to 1.5 T contrast.

| Method | MSE | AHD |
|---|---|---|
| *Original* | $1.85[\pm0.03] \times 10^{-3}$ | $6.15[\pm0.12] \times 10^{5}$ |
| *DH* | $2.30[\pm0.07] \times 10^{-3}$ | $7.49[\pm0.09] \times 10^{5}$ |
| *pGAN* | $\mathbf{1.50[\pm0.03] \times 10^{-3}}$ | $\mathbf{1.27[\pm0.07] \times 10^{5}}$ |
| *DM (ours)* | $1.61[\pm0.03] \times 10^{-3}$ | $1.31[\pm0.16] \times 10^{5}$ |

Table 7: MSE and AHD averages including variances over four-fold cross-validation for the mapping from 1.5 T to 3 T contrast.

| Method | MSE | AHD |
|---|---|---|
| *Original* | $1.85[\pm0.03] \times 10^{-3}$ | $6.15[\pm0.12] \times 10^{5}$ |
| *DH* | $1.73[\pm0.07] \times 10^{-3}$ | $8.22[\pm0.14] \times 10^{5}$ |
| *pGAN* | $0.94[\pm0.11] \times 10^{-3}$ | $2.07[\pm0.34] \times 10^{5}$ |
| *DM (ours)* | $\mathbf{0.76[\pm0.008] \times 10^{-3}}$ | $\mathbf{1.48[\pm0.16] \times 10^{5}}$ |

Table 8: Average CSF, GM and WM segmentation differences and variances over the four-fold cross-validation in mm$^3$. Differences between $\hat{B}$ and $B$ as well as between $\hat{B}$ and $B_{transformed}$ for the mapping from 3 T to 1.5 T contrast.

| Method | CSF | GM | WM |
|---|---|---|---|
| Original | $3.72[\pm 0.43] \times 10^4$ | $9.32[\pm 0.93] \times 10^4$ | $8.88[\pm 0.58] \times 10^4$ |
| pGAN | $\mathbf{1.12[\pm 0.17] \times 10^4}$ | $\mathbf{1.41[\pm 0.23] \times 10^4}$ | $1.91[\pm 0.19] \times 10^4$ |
| DM (ours) | $1.95[\pm 0.24] \times 10^4$ | $1.54[\pm 0.40] \times 10^4$ | $\mathbf{1.59[\pm 0.24] \times 10^4}$ |

Table 9: Average CSF, GM and WM segmentation differences and variances over the four-fold cross-validation in mm$^3$. Differences between $\hat{B}$ and $B$ as well as between $\hat{B}$ and $B_{transformed}$ for the mapping from 1.5 T to 3 T contrast.

| Method | CSF | GM | WM |
|---|---|---|---|
| Original | $3.72[\pm 0.43] \times 10^4$ | $9.32[\pm 0.93] \times 10^4$ | $8.88[\pm 0.58] \times 10^4$ |
| pGAN | $3.11[\pm 0.55] \times 10^4$ | $6.15[\pm 0.51] \times 10^4$ | $2.10[\pm 0.27] \times 10^4$ |
| DM (ours) | $\mathbf{1.79[\pm 0.06] \times 10^4}$ | $\mathbf{3.91[\pm 0.19] \times 10^4}$ | $\mathbf{1.49[\pm 0.65] \times 10^4}$ |

Table 10: Average Dice scores and variances over the four-fold cross-validation of segmentations of CSF, GM and WM of $B$ and of $B_{transformed}$ compared to ground truth segmentation of $\hat{B}$ for the mapping from 3 T to 1.5 T contrast.

| Method | CSF | GM | WM |
|---|---|---|---|
| Original | $0.8188[\pm 0.0095]$ | $\mathbf{0.8180[\pm 0.0041]}$ | $0.8715[\pm 0.0031]$ |
| pGAN | $\mathbf{0.8205[\pm 0.0075]}$ | $0.7925[\pm 0.0040]$ | $0.8745[\pm 0.0029]$ |
| DM (ours) | $0.7980[\pm 0.0052]$ | $0.7918[\pm 0.0029]$ | $\mathbf{0.8762[\pm 0.0027]}$ |

Table 11: Average Dice scores and variances over the four-fold cross-validation of segmentations of CSF, GM and WM of $B$ and of $B_{transformed}$ compared to ground truth segmentation of $\hat{B}$ for the mapping from 1.5 T to 3 T contrast.

| Method | CSF | GM | WM |
|---|---|---|---|
| Original | $\mathbf{0.8188[\pm 0.0095]}$ | $0.8180[\pm 0.0041]$ | $0.8715[\pm 0.0031]$ |
| pGAN | $0.7607[\pm 0.0113]$ | $0.8521[\pm 0.0049]$ | $0.9163[\pm 0.0034]$ |
| DM (ours) | $0.8094[\pm 0.0101]$ | $\mathbf{0.8760[\pm 0.0045]}$ | $\mathbf{0.9232[\pm 0.0031]}$ |

Table 12: Average Hausdorff distances and variances over the four-fold cross validation of segmentations of CSF, GM and WM of $B$ and of $B_{transformed}$ compared to ground truth segmentation of $\hat{B}$ for the mapping from 3 T to 1.5 T contrast.

| Method | CSF | GM | WM |
|---|---|---|---|
| *Original* | $10.21[\pm1.03]$ | $8.68[\pm0.83]$ | $11.16[\pm0.99]$ |
| *pGAN* | $21.48[\pm5.88]$ | $13.54[\pm3.53]$ | $12.14[\pm2.98]$ |
| *DM (ours)* | $\mathbf{9.11}[\pm\mathbf{1.05}]$ | $\mathbf{6.73}[\pm\mathbf{1.02}]$ | $\mathbf{7.67}[\pm\mathbf{1.02}]$ |

Table 13: Average Hausdorff distances and variances over the four-fold cross validation of segmentations of CSF, GM and WM of $B$ and of $B_{transformed}$ compared to ground truth segmentation of $\hat{B}$ for the mapping from 1.5 T to 3 T contrast.

| Method | CSF | GM | WM |
|---|---|---|---|
| *Original* | $10.21[\pm1.03]$ | $8.68[\pm0.83]$ | $11.16[\pm0.99]$ |
| *pGAN* | $8.67[\pm0.80]$ | $7.24[\pm0.90]$ | $9.48[\pm1.04]$ |
| *DM (ours)* | $\mathbf{8.65}[\pm\mathbf{0.91}]$ | $\mathbf{7.01}[\pm\mathbf{0.81}]$ | $\mathbf{9.30}[\pm\mathbf{1.33}]$ |

## Appendix E. Histogram Examples

In Figures 5, 6, 7 and 8 we show exemplary histograms of both, healthy control and MS patient scans for both directions of the image mapping. Each histogram consists of 255 bins. Each figure contains the histograms of $B$, $\hat{B}$ and the generated volumes $B_{transformed}$. The histograms of $DH$ are shifted towards the brightest voxels for both directions of the mapping. Due to this comparatively high amount of white voxels, the histograms of the $DH$ samples are cropped in Figures 5 - 8. The initial $1.5\,\mathrm{T}$ and $3\,\mathrm{T}$ histograms show big differences. Both show two peaks, but for the $3\,\mathrm{T}$ images, these are further apart, letting us perceive images of higher contrast compared to the $1.5\,\mathrm{T}$ images. In Figures 5 - 8, our $DM$ and $pGAN$ manage to align the histograms of the generated samples much closer with these of the ground truths $\hat{B} \in \mathcal{T}$ than the initial histograms of $B \in \mathcal{S}$. Therefore we assume that for both methods, the contrast harmonization is proving effective, for healthy control as well as for MS patient scans.

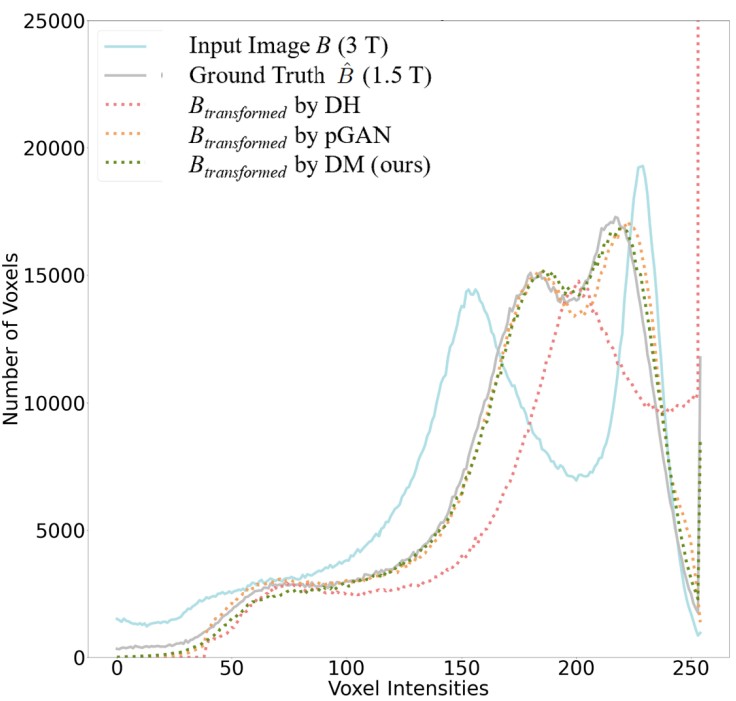

Figure 5: Exemplary histograms of one healthy control for the mapping from $3\,\mathrm{T}$ to $1.5\,\mathrm{T}$. The histograms of $B_{transformed}$ generated by $pGAN$ and our $DM$ align the ground truth histogram of $\hat{B}$ well compared to the histogram of the input image $B$.

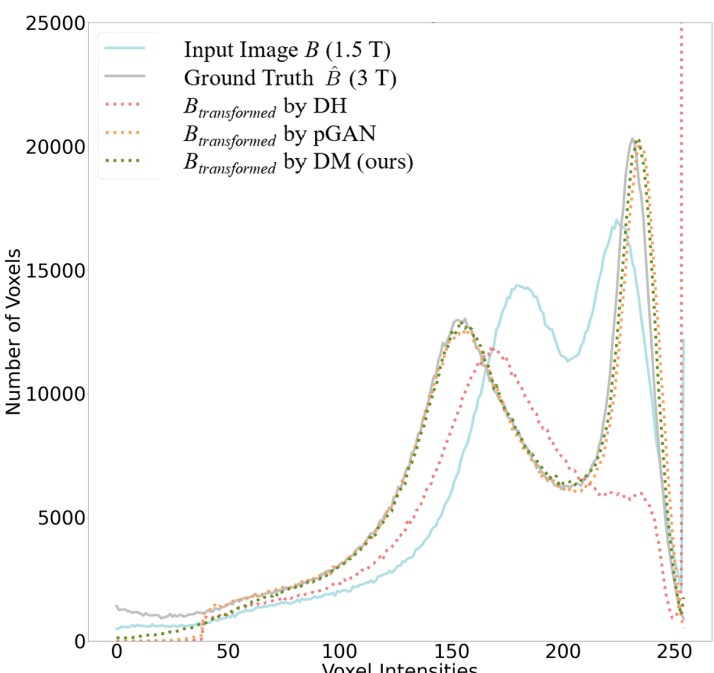

Figure 6: Exemplary histograms of one healthy control for the mapping from $1.5\,\mathrm{T}$ to $3\,\mathrm{T}$. The histograms of $B_{transformed}$ generated by $pGAN$ and our $DM$ align the ground truth histogram of $\hat{B}$ well compared to the histogram of the input image $B$.

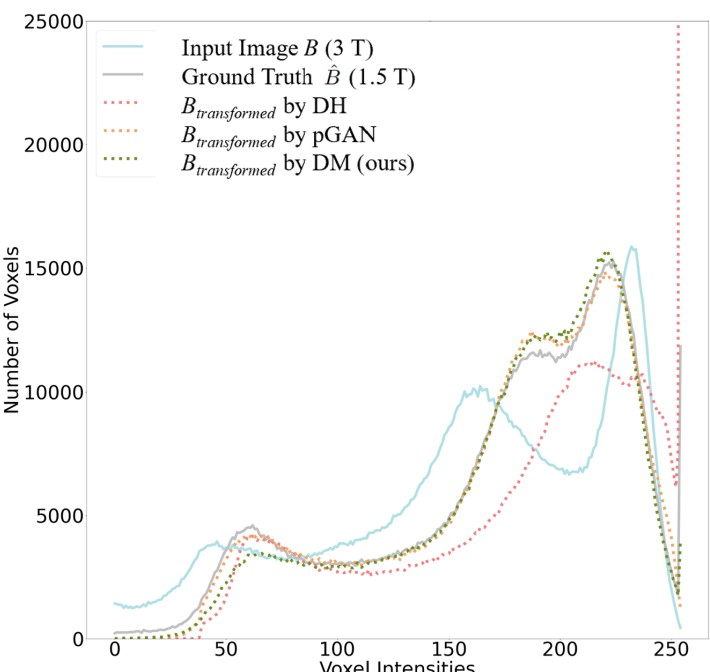

Figure 7: Exemplary histograms of one MS patient for the mapping from $3\,\mathrm{T}$ to $1.5\,\mathrm{T}$. The histograms of $B_{transformed}$ generated by $pGAN$ and our $DM$ align the ground truth histogram of $\hat{B}$ well compared to the histogram of the input image $B$.

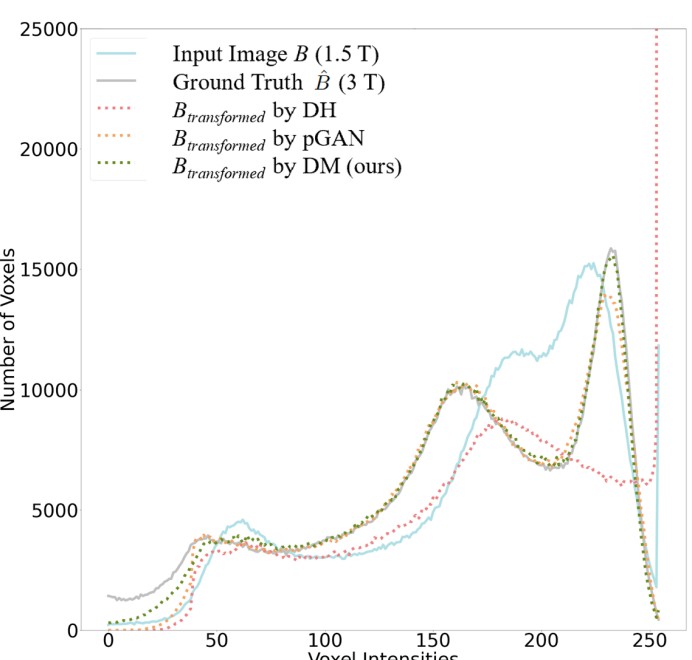

Figure 8: Exemplary histograms of one MS patient for the mapping from $1.5\,\mathrm{T}$ to $3\,\mathrm{T}$. The histogram of $B_{transformed}$ generated by $pGAN$ aligns the ground truth histogram of $\hat{B}$ well compared to the histogram of the input image $B$, while our $DM$ manages to almost perfectly match the peaks of $\hat{B}$.

## Appendix F. Image Examples

*3 T Original*
*B (Input)*

*1.5 T Original*
*$\hat{B}$ (GT)*

*$B_{transformed}$ by*
*DH*

*$B_{transformed}$ by*
*pGAN*

*$B_{transformed}$ by*
*our DM*

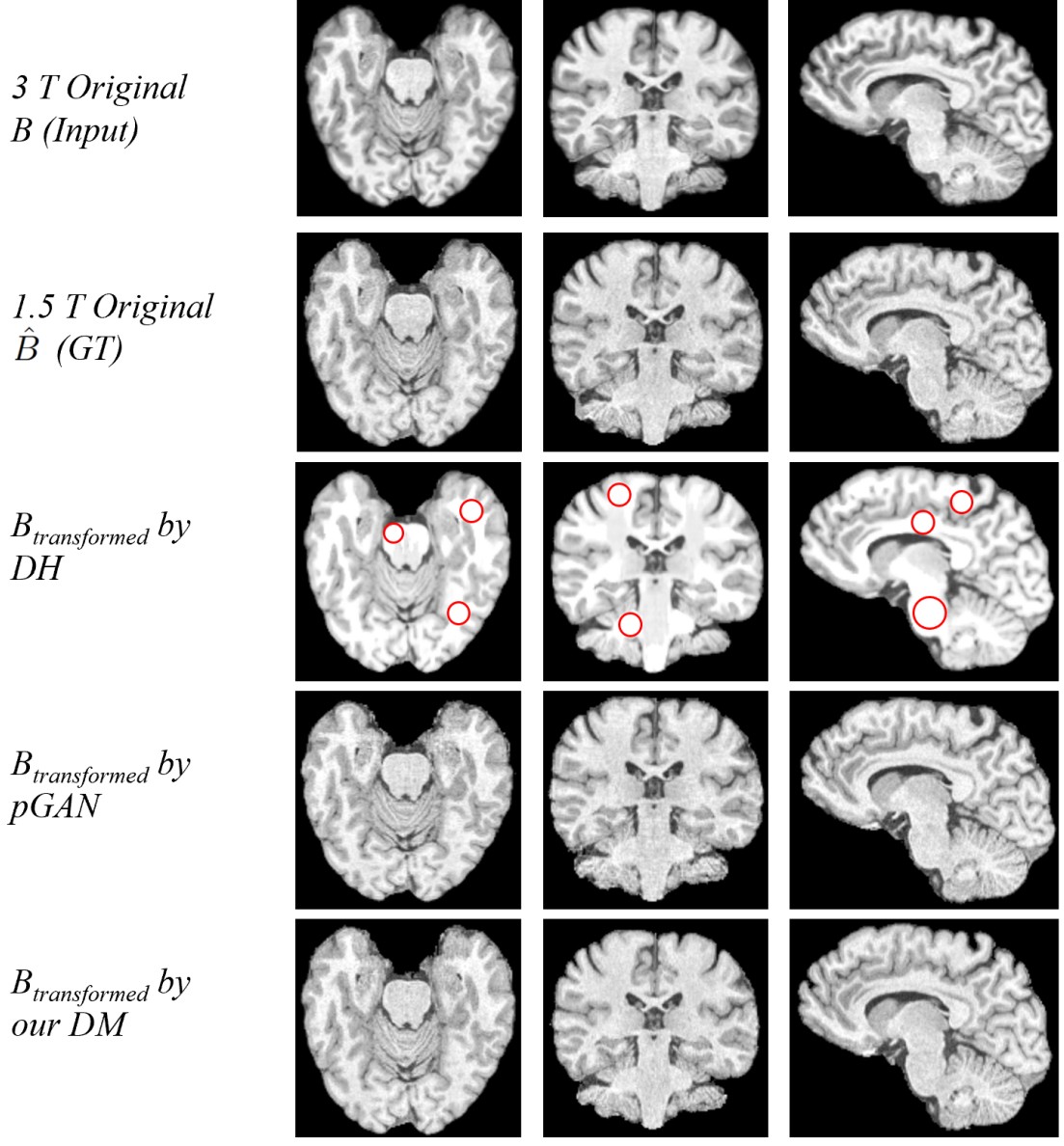

Figure 9: Exemplary images for the translation from $3\,\mathrm{T}$ source contrast to $1.5\,\mathrm{T}$ target contrast. The methods generate sagittal slices (right column), which are stacked to create a 3D volume. The red circles indicate hyperintense regions generated by *DH*.

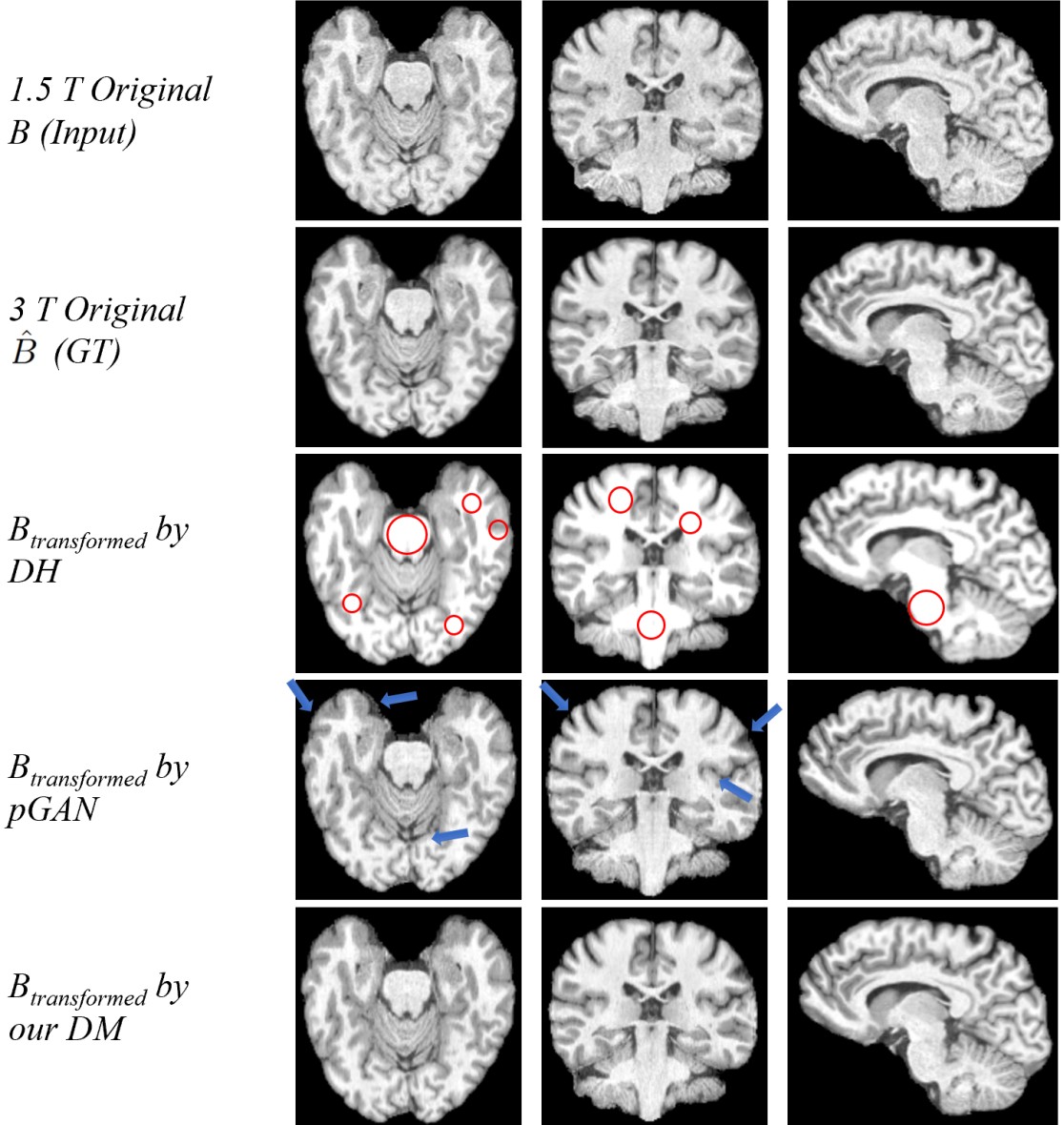

Figure 10: Exemplary images for the translation from 1.5 T source contrast to 3 T target contrast. The methods generate sagittal slices (right column), which are stacked to create a 3D volume. The red circles indicate hyperintense regions generated by *DH*. The blue arrows point at stripe artifacts due to the stacking of 2D slices produced by the *pGAN*.

## Appendix G. Segmentation Examples

*Segmentation of*
*3 T Original B (Input)*

*Segmentation of*
*1.5 T Original $\hat{B}$ (GT)*

*Segmentation of*
*$B_{transformed}$ by pGAN*

*Segmentation of*
*$B_{transformed}$ by our DM*

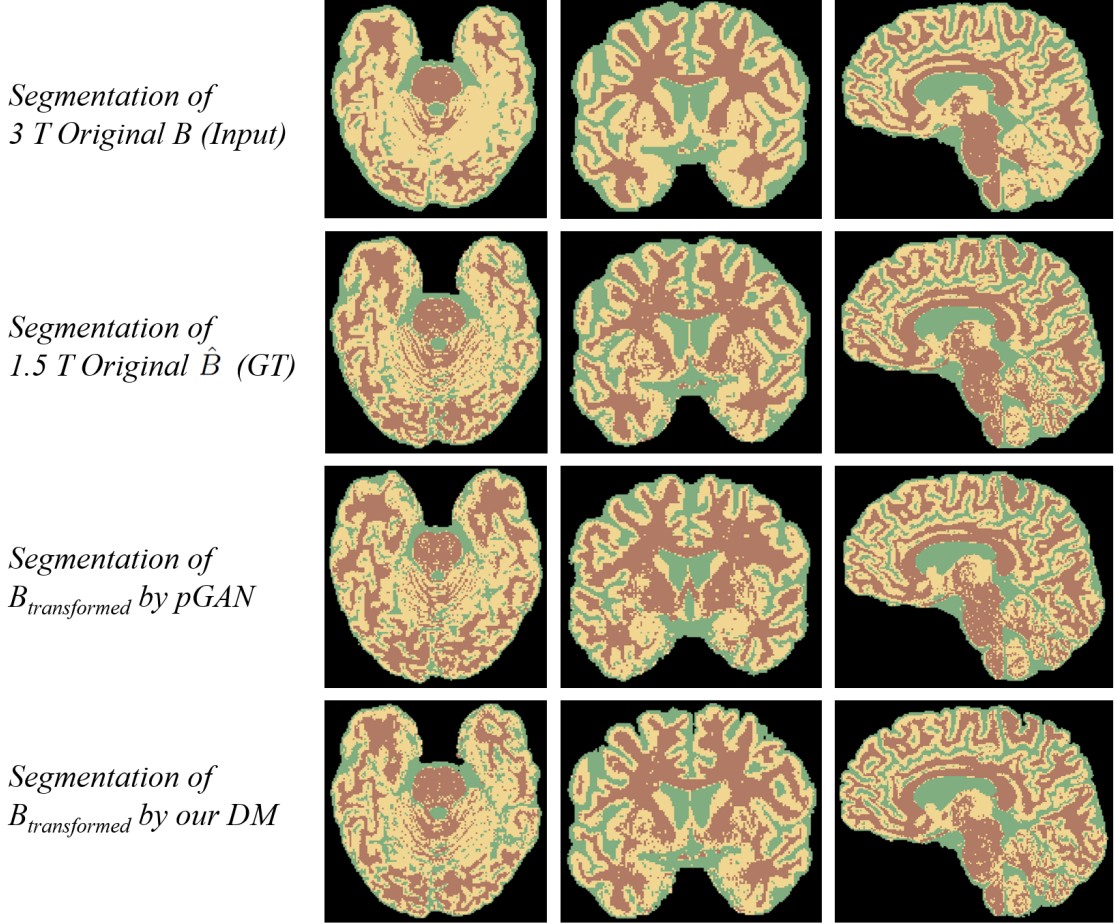

Figure 11: Exemplary segmentations for the translation from 3 T source contrast to 1.5 T target contrast. Green indicates CSF, yellow is GM and red is WM.

*Segmentation of
1.5 T Original B (Input)*

*Segmentation of
3 T Original $\hat{B}$ (GT)*

*Segmentation of
$B_{transformed}$ by pGAN*

*Segmentation of
$B_{transformed}$ by our DM*

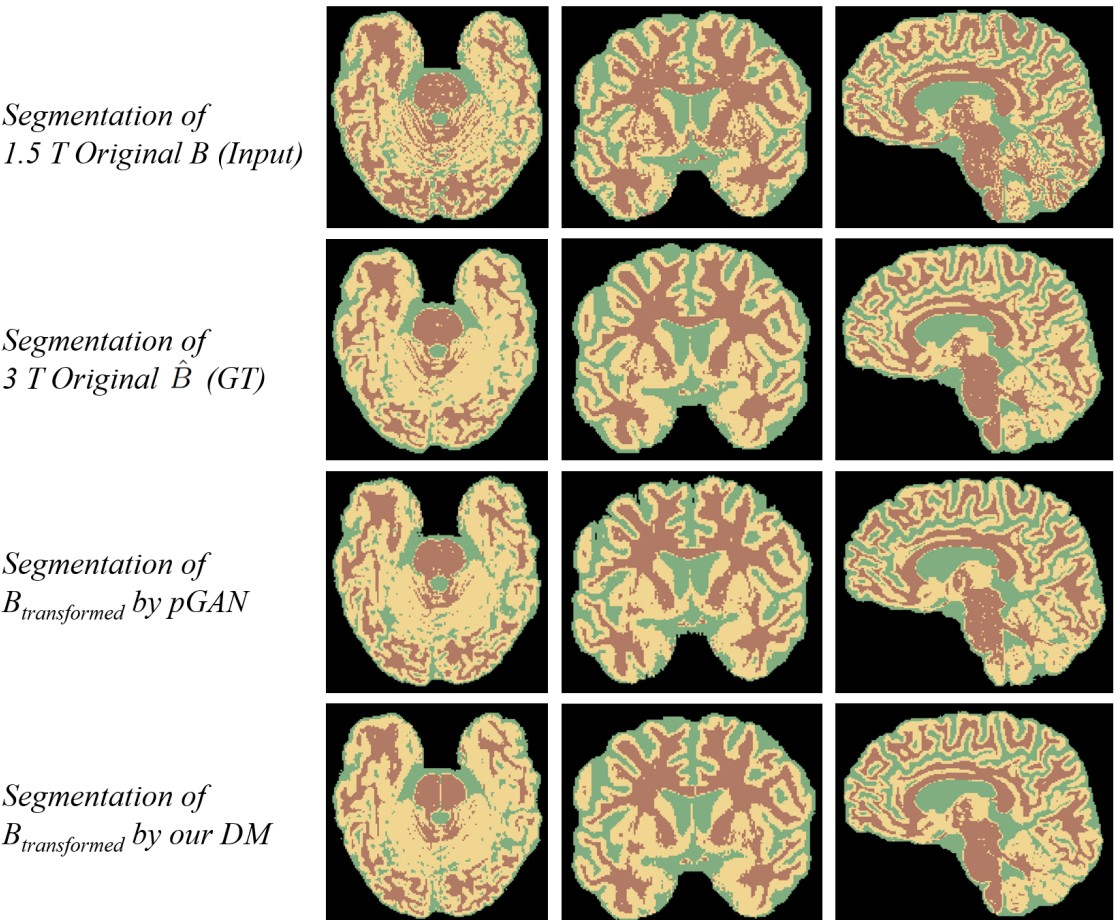

Figure 12: Exemplary segmentations for the translation from 1.5 T source contrast to 3 T target contrast. Green indicates CSF, yellow is GM and red is WM.

