# OpenReview forum: "Diffusion Models for Contrast Harmonization of Magnetic Resonance Images"
_MIDL.io/2023/Conference — MIDL 2023 Poster_

### Official Review · Reviewer_w4tZ · 2023-02-04

**Confidence:** 4
**Preliminary Rating:** 4
**Recommendation:** Poster

**Summary:**

The paper presents a diffusion model-based approach to harmonize contrast between MRI images taken with 1.5 T and 3 T scanners. The algorithm needs paired images of the same subject. In the experiments, they validated the technique by mimicking the 1.5 T contrast using 3 T images as input and vice versa. The results show improved performance compared to other related methods, especially when mapping from 1.5 T to 3 T.

**Strengths:**

MRI images can differ significantly when captured using different scanners or settings, making it challenging to compare images taken under varying conditions. As such, contrast harmonization is a crucial preprocessing step for analyzing inhomogeneous datasets. The paper uses diffusion models to perform the contrast harmonization and, to maintain anatomical features, the authors propose incorporating data from the baseline image. The experiments show that for the specific dataset, the methods successfully changed the contrast while preserving anatomical features.

**Weaknesses:**

The experiments were based on a single dataset with a limited number of subjects, which may limit the generalizability of the findings. It is important to consider the impact of this method on clinically relevant conclusions derived from data analysis using different techniques. The current experiment appears to be insufficient in this regard and further exploration is needed. It would be valuable to evaluate the method on a dataset containing subjects with tumors, stroke lesions, or white matter hyperintensities.

**Deanonymize Review:**

no

**Detailed Comments:**

The last paragraph in section 2 beginning with "For the two directions of the transformation, 1.5 T to 3 T and 3 T to 1.5 T, two separate
models need to be trained, as source and target contrast change. .." : It appears a bit out of the blue (3 T to 1.5 T).

Regarding Eq. (5), it is possible that a small constant should be added in front of ε. The equation could be written as |k ε - blabla|, where k is the constant. Please verify this with the relevant sources.

**Paper Type:**

methodological development

**Questions To Address In The Rebuttal:**

It would be valuable to conduct an experiment using another dataset, as it would provide additional insights into the effectiveness of the proposed method. While obtaining a set of paired images may be challenging, one option could be to artificially downgrade images or simulate MRI distortions. This could provide further insights into the robustness of the proposed method.

---

### Official Review · Reviewer_2TzZ · 2023-02-04

**Confidence:** 3
**Preliminary Rating:** 4

**Summary:**

This paper addresses the problem of image harmonization between different MRI scanners / field strengths (here 1.5T and 3T).
A diffusion model-based approach is considered, translating the contrast only but not the anatomical information. The method is evaluated on paired datasets assessing the segmentations of CSF, GM and WM.

**Strengths:**

The paper addresses an important problem and uses a method which has not been used in this exact way for this problem. It choses a sensible evaluation strategy on a nice paired dataset and compares adequately with state of the art. The introduction and especially the state of the art section are comprehensive and include a range of very recent publications.

**Weaknesses:**

The paper could be clearer especially in the evaluation of the results and the discplay of the figures. Furthermore, the abstract needs to be improved, making esp. clearer what type of data, how many datasets were used and giving a sense of how well it quantitatively performed would be important.

**Deanonymize Review:**

no

**Detailed Comments:**

-abstract: no detail on how many datasets, information on performance etc!
-Introduction well written, good amount of previous publications highlighted, concrete use could be clearer
-The authors highlight that the short time span between images for the MS patients signifies that no major progression has occured. I would nevertheless be critical that this algorithm is trained to a large part on MS data - maybe the different lesions affect the different contrast changes between 1.5T and 3T in a particular manner? Can the authors please comment on this?
-any evaluation on another dataset would be very helpful, can the trained network not just be applied to another open source dataset for this purpose, eg to a 3T dataset, volumes of GM, WM and CSF measured and compared to normal ranges from other 1.5T datasets? Maybe not that easy, but the dataset used and the balance MS / HC seems a real limitation here, esp given how much MRI data is out there any publicly available!
-Figure 4 needs to be clearer, eg zooms or arrows to appreciate the differences better.
-Table 1: too many abbreviations, there is space, you can spell out the methods names and/or measure names at least in the caption.
-Figure 5: font to small in the legend
-Table 3: this is a strange table with only one row and one heading row?
-Figure 6: looks too bright, similar to Figure 4 should have zooms /arrows etc!
(same for 7,8 and 9 ; ))
-Conclusion: in the part comparing to GANs, the authors say this "mak[es] the training straightforward." - sure, but can you qwuantify a bit? eg how much faster, requirements on GPU etc reduced by how much?
- NOt clear to me why skull stripping was applied in the first place? the authors say omitting this is the next step, but why is it currently done at all?

**Paper Type:**

methodological development

**Questions To Address In The Rebuttal:**

The authors should try to make their paper more concise and clearer esp. in the results section. Highlight clearer using arrows or zoooms etc what is better in which way in the new method and why this could be the case. Also more details on teh acquisition side would be nice, esp things like which preprocessing (vendor-supplied filtering for example) was applied. Finally, the authors shoudl address why they thing the dataset being made up to a large extent from MS patients is not a limitation.

---

### Official Review · Reviewer_ekJG · 2023-02-05

**Confidence:** 5
**Preliminary Rating:** 3
**Recommendation:** Poster

**Summary:**

this study proposes to use diffusion models, which is recently hot topic in ML community, to harmonize MRI scans (contrasts). Authors trained a model to transform source domain to target domain, where source and target domains can be considered different scanners. This is having the same notion of image-to-image translation. Structural information (from segmentation) are used to enhance the models.

**Strengths:**

-- The use of diffusion in MRI harmonization seems to be novel application.
-- Code is available, the study maybe reproducible (not checked by this reviewer yet).
-- comparisons are made with at least two methods.
-- results are on par with other methods.

**Weaknesses:**

-- vague descriptions exist. What is "better compatibility", how do we measure compatibility ?
-- it needs to write better and more clear what the contribution is, maybe a separate subsection
-- figures are small, hard to understand and read
-- results are not better than pGAN for 3T to 1.5T, what is the reason while the opposite is not true (from 1.5T to 3T)?
-- what does figure 5 tell us?
-- dice scores are high, not clear if DM works better than others, there is no standard deviation and Hausdorff distance metrics.


**Deanonymize Review:**

no

**Detailed Comments:**

please see weaknesses comments, these are self-explanatory question based comments.

**Paper Type:**

validation/application paper

**Questions To Address In The Rebuttal:**

-- vague descriptions exist. What is "better compatibility", how do we measure compatibility ?
-- it needs to write better and more clear what the contribution is, maybe a separate subsection
-- figures are small, hard to understand and read
-- results are not better than pGAN for 3T to 1.5T, what is the reason while the opposite is not true (from 1.5T to 3T)?
-- what does figure 5 tell us?
-- dice scores are high, not clear if DM works better than others, there is no standard deviation and Hausdorff distance metrics.

---

### Meta-Review · Area_Chair_qVdj · 2023-02-24

**Recommendation:** Accept (Poster)
**Confidence:** 5

**Metareview:**

The paper proposed to use the improved DDPM model to harmonize MRI scans.  While initial reviews had several concerns regarding lack of technique novelty and missing details/vague descriptions, the author addressed these comments in the rebuttal. After rebuttal, the paper received 2 weak accepts and 1 broadline, which met the requirement for publication.